## PERSPECTIVE

# Guiding principles for the responsible development of artificial intelligence tools for healthcare

Kimberly Badal [1✉], Carmen M. Lee [2] & Laura J. Esserman [1]

Several principles have been proposed to improve use of artificial intelligence (AI) in healthcare, but the need for AI to improve longstanding healthcare challenges has not been sufficiently emphasized. We propose that AI should be designed to alleviate health disparities, report clinically meaningful outcomes, reduce overdiagnosis and overtreatment, have high healthcare value, consider biographical drivers of health, be easily tailored to the local population, promote a learning healthcare system, and facilitate shared decision-making. These principles are illustrated by examples from breast cancer research and we provide questions that can be used by AI developers when applying each principle to their work.

Artificial intelligence (AI) is projected to have a transformative impact on clinical medicine, biomedical research, public and global health, and healthcare administration[1,2]. The enthusiasm for AI applications in healthcare is especially evident in the United States of America (USA), where, as of September 2021, there were 343 Artificial Intelligence/Machine Learning-enabled medical devices approved for use by the Food and Drug Administration (FDA), with the vast majority in radiology[3]. The immense interest in the application of artificial intelligence (AI) in healthcare drove the development of AI principles by policy, regulatory, and professional organizations, including the FDA[3], Health Canada[3], the World Health Organization (WHO)[4], and the American Medical Informatics Association (AMIA)[5] (Table 1). There is considerable synergy between these proposed AI principles, demonstrating an evolving global consensus on what constitutes responsible AI for healthcare.

We propose that the published guiding AI principles should be expanded. Firstly, the principles do not explicitly require that AI tools are intentionally designed to contribute to fixing deeply engrained and too often overlooked challenges in healthcare, a requirement we see as essential. Without explicit attention to these issues, AI will not improve healthcare but will lead to more tools that reinforce pre-existing systemic challenges. Secondly, the published principles are often written for a broad, multi-stakeholder audience rather than explicitly for the AI developer who is ultimately responsible for model development. Given the finite time and resources allocated to AI development and that AI developer teams often lack access to specialized multi-stakeholder expertise, it is imperative that AI developers are provided with a clear, thorough, and systematic way to integrate the proposed principles into AI development. The FUTURE-AI Medical AI Algorithm Checklist[6] is an example of a checklist framework that translates high-level AI principles into practical computational guidance. The proposed TRIPOD-AI and PROBAST-AI checklists will provide guidance on how to report and critically appraise AI models developed for diagnosis or prognosis[7]. Without such assistive checklists, it will become increasingly difficult for AI developer teams to action principles in computer sciences and in the various other domains that AI traverses, such as clinical medicine, biomedicine, ethics, and law.

This perspective offers eight principles that we believe must be addressed when developing AI tools for healthcare (Table 2). We focus on the computational scientist as the primary audience and emphasize that AI must be purposely designed to improve longstanding, systemic

---

[1] Department of Surgery, Helen Diller Comprehensive Cancer Center, University of California, San Francisco, CA, USA. [2] Department of Emergency Medicine, Highland Hospital, Alameda Health System, Alameda, CA, USA. ✉email: kimberly.badal@ucsf.edu

**Table 1 Summary of AI principles proposed by select organizations.**

| Policy or framework | Organization | Principles | Primary audience |
|---|---|---|---|
| Ethics and governance of artificial intelligence for health | World Health Organization | • Human autonomy<br>• Human well-being and safety and the public interest<br>• Transparency, explainability, and intelligibility<br>• Responsibility and accountability<br>• Inclusiveness and equity<br>• Responsive and sustainable | Ministries of Health |
| Medical AI algorithm assessment checklist | FUTURE-AI (an international, multi-stakeholder consortium) | • Fairness<br>• Universality<br>• Traceability<br>• Usability<br>• Robustness<br>• Explainability | AI teams |
| Good Machine Learning Practice for Medical Device Development: Guiding Principles | FDA, Health Canada, United Kingdom's Medicines and Healthcare products Regulatory Agency (MHRA) | • Leverage multidisciplinary expertise in development<br>• Implement good software engineering and security practices<br>• Datasets are representative of intended population<br>• Training and test sets are independent<br>• Reference datasets are well developed<br>• Optimize performance of Human-AI Team<br>• Thorough clinical testing<br>• Information accessible to users<br>• Monitor deployed models and mitigate retraining risks | AI medical device developers |
| Defining AMIA's artificial intelligence principles[a] | American Medical Informatics Association (AMIA) | • Autonomy<br>• Beneficence<br>• Non-maleficence<br>• Justice<br>• Explainability<br>• Interpretability<br>• Fairness<br>• Dependability<br>• Auditability<br>• Knowledge management | All stakeholders affected by or involved with an AI system |

[a]Three additional principles for organizations developing or deploying AI and three special considerations not included.

challenges in healthcare. We use examples from breast cancer research to illustrate why these principles are important. We also frame questions to enable AI developers to probe these principles (Table 2). Some principles overlap with existing work, while others, to our knowledge, have not been explicitly explored. The eight principles are not exhaustive; they should be integrated with other work, tailored to the intended AI application, and improved over time. While many of these principles could be applied to any health technology, we focus this perspective on AI because the nuance of this technology lends itself to unique considerations (e.g., principle 6) and opportunities (e.g., principles 5 and 7) and for comparison to existing AI policies and frameworks (Table 1).

## Principle 1: AI tools should aim to alleviate existing health disparities

Moving global health equity forward is long overdue. Health equity means reducing and ultimately eliminating the disparities in health outcomes that exist between advantaged and disadvantaged populations caused by the disproportionate exposure

of disadvantaged groups to risk factors and poor access to high-quality care. AI tools will likely only realize benefits in populations that already benefit the most from healthcare, thus widening the health equity gap. This is because AI tools usually require the collection of specialized data for inputs, cloud or local computing for hosting, high purchasing power for acquisition from commercial companies, and technical expertise, all of which are barriers to entry into hospital systems that serve the most disadvantaged populations.

AI tools should not introduce, sustain, or worsen health disparities but must instead be intentionally designed to reduce known disparities if there is to be tangible progress toward health equity. The proposed principles of inclusiveness, fairness, and equity (Table 1) all capture a desire to address health disparities. There is also a growing body of literature that discusses how AI can be used to address health disparities[8–10]. For illustration, we focus on two practical strategies, which are to ensure that disadvantaged groups can equally access and benefit from the AI tool and to preferentially design the AI tool for disadvantaged groups.

**Table 2 Questions that can be used when considering each principle in the AI development process.**

| Principle | Questions |
|---|---|
| 1. Alleviate healthcare disparities | • What health disparities are reported for the present AI application?<br>• How can the AI tool be designed to be accessible to and improve outcomes for the disadvantaged population?<br>• What clinical interventions are needed to realize the benefit, and are these accessible?<br>• How can data collection be supported in underserved communities for tool retraining over time? |
| 2. Report clinically meaningful outcomes | • How is clinical benefit defined in this domain?<br>• What is the present threshold for the clinical benefit of existing tools, and how can the AI tool improve upon this threshold? |
| 3. Reduce overdiagnosis and overtreatment | • What disease state is an overdiagnosis?<br>• For every case of overdiagnosis, what are the downstream costs to the patient and healthcare system?<br>• How can this AI application reduce the number of overdiagnoses compared to existing approaches? |
| 4. Have high healthcare value | • Is this AI tool addressing a high-priority healthcare need?<br>• What would be the cost to the healthcare system in implementation, maintenance, and update?<br>• What would be the cost to the patient who does and does not benefit from this tool?<br>• Does this tool have high healthcare value, and if not, how can it be improved? |
| 5. Incorporate biography | • What biographical data can be collected or carefully coded for the intended population?<br>• How do these factors vary in the intended population?<br>• How can these factors be included when developing AI tools? |
| 6. Be easily tailored to the local population | • Can the training features be easily collected in different settings?<br>• Are these features reliable for training across different populations?<br>• Will the AI/ML workflow be made open-access? |
| 7. Promote a learning healthcare system | • How will this AI application be evaluated over time, and at what intervals?<br>• What are acceptable thresholds for performance?<br>• How will the evaluation results contribute to continuous improvement? |
| 8. Facilitate shared decision-making | • Have AI explainability tools been explored and utilized?<br>• Do clinicians and patients find the explainability results helpful?<br>• Have simpler, explainable algorithms been tried and compared to 'black-box' algorithms to determine if a simpler model performs just as well?<br>• How can patient values be easily integrated into the use of the AI tool? |

The first strategy of ensuring equal access and benefit can be challenging. For example, African American (AA) breast cancer patients in the US have higher mortality rates relative to white women, which is attributed to disparities in access to screening and endocrine therapy[11]. An AI tool for breast cancer screening (e.g., AI tools that predict breast cancer risk) intentionally designed to ensure that AAs have equal access and benefit would require both training on datasets with balanced, unbiased representation of AA populations and a design that is accessible to and works for hospitals that serve AAs. Concrete steps to mitigate the systemic biases entrenched in the US healthcare system, and therefore present in training datasets, is explored in the literature[12,13].

AAs often live in areas with low access to primary care physicians[14] and are often served by hospitals with low resources[15] and poor care quality[16]. Therefore, ensuring AI tools work in these settings may require developers to prioritize the use of routinely collected or inexpensive data points as inputs, prioritize the use of single, explainable algorithms that can be run on a local computer, and advocate for commercial companies to provide discounted products, free cloud access, and the local training required for AI maintenance. Thus, creating an equitable AI tool may require prioritizing 'simpler' models for deployment therefore, in some instances, performance may be sacrificed. However, we must remember that the collective investment in resources and effort used to create AI tools must endeavor to benefit all rather than the few. The trade-off between balancing accuracy and equity can potentially be resolved by designing AI tools that can be easily tailored to the local population (principle 6).

The second strategy to reduce the disparity in breast cancer mortality rates would be to prioritize developing AI tools for AA-serving hospitals over other hospitals. This strategy is essentially a form of affirmative action in healthcare[17]. In the USA, affirmative action refers to policies that aim to increase the representation of minorities or address the disadvantages they suffer[17]. The application of affirmative action policies to AI development will require careful evaluation of the ethical implications. Do advantaged groups who will not have access to the AI tool miss an immediate opportunity for improved outcomes? Is this missed opportunity ethically justifiable? Given that AAs are more likely to die from breast cancer, prioritizing developing tools to reduce AA mortality rates could be considered to be ethically justifiable in the same way that those at the highest risk of death during the COVID-19 pandemic were prioritized for vaccination[17]. However, this strategy will be ineffective if AA populations do not have access to the screening or risk-reducing interventions recommended by the AI tool or access to therapeutic interventions once diagnosed. Therefore, a combination of need and capacity to benefit is often needed to justify preferential resource allocation[17]. AI tools designed to serve disadvantaged groups must have the potential to be materially beneficial, given the healthcare system's limitations. If not, the tool will likely have low healthcare value and will unnecessarily divert resources from higher priority areas and more effective interventions (principle 4).

## Principle 2: Outcomes of AI tools should be clinically meaningful

The field of clinical medicine has evolved over decades of thoughtful research and intervention. In many diseases, there are clinical outcomes that are agreed upon as a metric of successful intervention. These outcomes change over time as the collective understanding of disease progresses. In breast cancer screening, for example, the number of stage 0 tumors detected was a measure of success until it was realized that many of these tumors did

not progress to be clinically meaningful and that the increase in the detection of stage 0 or in situ tumors was not accompanied by a concomitant decrease in invasive cancers[18,19]. The field of breast cancer screening has since evolved to consider other short- and long-term metrics of success, such as the number of late stage or interval (i.e., found between mammography screens) cancers averted or the number of deaths averted[20]. If AI researchers do not define clinical benefit from the start, they risk creating a tool clinicians cannot evaluate and use. Clinicians need to evaluate the accuracy, fairness, risks of overdiagnosis and overtreatment (principle 3), healthcare value (principle 4), and the explainability, interpretability, and auditability (Table 1) of AI tools. Such evaluations are difficult with tools that do not predict clinically meaningful outcomes. Further, identifying the type of benefit desired from the outset avoids the development of tools that inadvertently find disease that leads to overtreatment (principle 3). It should be noted that in some domains, it may be difficult to define clinical benefit; however, this does not preclude the need to identify an acceptable definition of benefit.

### Principle 3: AI tools should aim to reduce overdiagnosis and overtreatment

Overdiagnosis and overtreatment are often viewed as acceptable costs of correctly diagnosing all disease instances, that is, favoring sensitivity over specificity. This is because of the high value placed on the potential of correct medical intervention. However, the physical, emotional, and financial costs of overdiagnosis and the overtreatment of patients must be considered. This is challenging because the definition of overdiagnosis is not always agreed upon, and definitions shift with the evolving understanding of the spectrum of disease. For example, ductal carcinoma in situ (DCIS) breast tumors sometimes remain indolent, meaning that it does not progress to invasive breast cancer, while some do progress. Therefore, in some cases, DCIS could be considered an overdiagnosis of breast cancer[21]. There are also invasive breast tumors that have a very low risk of recurrence[22]. Does identifying high-risk for progression DCIS cases and very low-risk for recurrence invasive cancer cases also constitute an overdiagnosis? Some of the AI tools designed to predict breast cancer risk do not differentiate between DCIS and invasive cancer[23,24]. This means that these tools will likely maintain or, in the worst-case scenario, exacerbate the rate of overdiagnosis and overtreatment. A better strategy would be to develop AI tools that predict subtype-specific breast cancer risk. Such a tool can be used to appropriately tailor interventions according to the predicted disease severity, thereby reducing overdiagnosis and overtreatment.

### Principle 4: AI tools should aspire to have high healthcare value and avoid diverting resources from higher-priority areas

Healthcare value is defined as the health outcomes achieved per dollar spent[25]. AI tools should increase healthcare value, meaning that they should provide better outcomes for the same cost as existing tools or the same outcome for less cost. The cost to gather inputs, implement, maintain, update, interpret, and deliver the results and the immediate and downstream cost of errors must be estimated. It is not enough to have a good working tool, it must make financial sense to the healthcare system and not increase costs for patients. An initial consultation at the outset with leadership stakeholders and health economists can establish whether and how the AI tools should or could be a financial priority. Furthermore, estimating the value of the tool benchmarked against the existing practice is imperative. Low-priority, low-value AI tools will divert resources from more critical areas. For

example, the present breast cancer screening paradigm in the US is expensive[26], and AI tools for screening should aim to reduce the cost to the health system and to the patient while increasing benefit[27].

This principle is particularly important in settings where scarce resources could be wasted on AI tools that will not have the same impact as other proven, foundational interventions. In such low-resource settings, it may not be feasible to assess healthcare value due to the absence of the requisite technical expertise. In such cases, a holistic view of the capacity of the healthcare system to realize AI benefits is needed. For example, in 2013, the WHO outlined why organized breast cancer screening programs should not be a priority in limited-resource settings with relatively strong or weak health systems[28]. One reason is the lack of organizational and financial resources necessary to sustain a screening program. Another reason is that screening benefits would not be realized if the healthcare system cannot provide adequate treatment and management for diagnosed patients[28]. The same arguments apply to prioritizing the development and deployment of AI tools for breast cancer screening in low-resource settings.

### Principle 5: AI tools should consider the biographical drivers of health

Accumulating evidence across disease states suggests that biological mechanisms alone cannot explain the disease. The biology of disease onset and progression can be impacted by a person's biography, that is, their lived experience. Biography is a newer conceptual field of research that comprises more than social determinants of health[29]. Biography is broadly conceived as a person's social, structural, and environmental exposures and affective emotional states[29–31]. Examples include allostatic load (cumulative burden of chronic stress and life events)[32], access to care, depression, and environmental pollution. Geographers have proposed conceptual frameworks for investigating how the body interacts with the environment[33], but its integration into medical research has not yet been realized, partly due to the lack of an overarching scientific discipline that equally investigates both biography and biology in understanding disease[30]. AI tools will miss the goal of delivering precision medicine interventions if the biographical drivers of health that contribute to the variation in outcomes seen between patients are not seriously considered. Importantly, machine learning is likely to be a key tool that will help uncover the complex relationships between biology and biography. At first, AI developers can utilize low-resolution data that could provide biographical information such as zip code and socioeconomic status scales until higher resolution, individualized biographical features can be collected. Biographical data can also be enriched by using zip codes to geocode neighborhood characterizations, exposures to environmental toxins, and other geospatial information, for example[34,35]. Essentially, deliberate thought and effort should be placed in determining how biographical determinants of health can be integrated into AI tools, with the goal of improving the resolution of these variables over time.

### Principle 6: AI tools should be designed to be easily tailored to the local population

AI researchers often seek external datasets as a test set to evaluate whether the tool is generalizable. These external datasets are often sourced from similar, high-resource settings, such as academic hospitals that serve relatively homogenous populations. This practice only demonstrates very limited generalizability to the populations similar to the test set. The highest form of generalizability in the global sense across populations, healthcare systems, and over time is likely impossible to attain and undesirable

given that generalizability occurs at the expense of precision, that is, the bias-variance tradeoff. The myth of generalizability in healthcare has been previously explored[36]. Poor generalizability is not unique to AI and is also a challenge with traditional statistical models. For example, breast cancer polygenic risk scores developed on women of European ancestry do not generalize well to people of African ancestry[37]. Similar trends have been noted in other diseases[38].

Rather than broad goals of generalizability, AI tools can instead be designed to be easily trained to maximize precision in the local population. This could mean using inputs that are easily collected and reliable training features across different populations such that algorithms can be retrained for a specific setting. Another strategy is to openly publish AI workflows or to provide platforms that institutions can use to train and evaluate their own local models.

## Principle 7: AI tools should promote a learning healthcare system

A major asset of AI is that continuous learning is necessary to ensure optimization and resilience over time from known challenges such as dataset shifts and noise[39]. The FDA is also investigating whether companies should be allowed to submit a 'change control plan' that will allow for changes to approved AI software while in deployment[40]. We view this as a matter of necessity, not an option. All interventions, AI or not, should be designed with the intention of regular evaluation, learning, and improvement[27]. One reason is that there are many opportunities for unexpected errors in AI deployment. Further, as science evolves, there should be mechanisms to integrate new knowledge that could benefit the patient. Evaluation metrics, timeframes, and performance standards should be determined in the AI research phase in consultation with clinicians. This evaluation must include not only global performance metrics such as specificity but also a granular understanding of who the tool does not work for, why it does not work, what the impact is on the patient and healthcare system, and provide a framework for improvement. This requirement overlaps with the principles of robustness and dependability (Table 1). An example of an AI monitoring and improvement framework was proposed by Feng et al where the authors explain how existing hospital quality assurance and improvement tools can be adapted to monitor ML algorithms[41].

## Principle 8: AI tools should facilitate shared decision-making

Some machine learning algorithms—specifically 'black-box' deep learning algorithms—are difficult to explain and interpret. The need for AI tools to be explainable (the internal logic of the system can be understood) and interpretable (the cause of a decision can be understood) has been consistently recognized as a central principle by many organizations (Table 1). Opaque AI tools cannot be adequately evaluated and audited, undermine trust[42,43], and cannot facilitate shared, informed decision-making between patient and practitioner[44]. Shared decision-making means that the patient is provided with adequate information about the intervention, which is considered along with their preferences and values (e.g., belief systems and risk tolerance) as a decision is made[44]. This is challenging if the patient and practitioner do not understand how and why the AI tool arrived at a decision[44]. An example is the case of offering patients diagnosed with DCIS either surgery or active surveillance. To facilitate this decision, the patient and practitioner would need to understand the risks and benefits of each option. If an AI tool is designed to assist with this decision, the patient and practitioner would also need to know how and why the recommendation was made and the advantages and limitations of the AI tool.

To ensure that AI tools make patient understanding and values central, AI researchers can utilize different explainability tools[45] and prioritize simpler, more intuitive algorithms. Another method recommended by Birch et al is to have AI risk prediction tools generate a continuous score rather than a fixed score so that decision threshold determination can be left to the patient and physician based on risk preferences[46].

## Conclusion

The collective innovation concentrated on AI applications in health must be guided to ensure that AI tools intentionally contribute to addressing longstanding shortcomings in healthcare. Doing so requires the thoughtful and systematic integration of principles that traverse many disciplines, which can be a daunting task for the AI developer. Clear and comprehensive guidance written explicitly for the developer, as presented here, is critically needed if the proposed principles are to be actioned. The eight principles outlined, in conjunction with those already proposed, will raise the standard to which AI tools are held. We do not see these principles as optional but critical and overdue to realize the promise of AI benefits in healthcare.

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

## Acknowledgements

We would like to thank Ralph Horwitz, Ida Sim, Allison Hayes-Conroy, Mark Cullen, and Burton Singer for their conversations on integrating biography into the study and practice of medicine.

## Author contributions

K.B. conceptualized, wrote, and revised the paper. C.L. reviewed the paper. L.J.E. substantially reviewed the paper.

## Competing interests

The authors declare no competing interests.
