## [Peer Review File · Communications Medicine]

Reviewers' comments:

Reviewer #1 (Remarks to the Author):

Preamble

1. Thank you for the opportunity to review your principles for AI. You have entered a crowded field. Ted Shortliffe and colleagues are about to publish an entire volume on cognitive computation in biomedicine, and at least two professional societies, the AMA and AMIA, also offer definitions and principles, in addition to the organizations you mention. A little further afield, the IEEE has several standards working groups with remits that at least overlap medicine and healthcare.
2. It is interesting to see to what extent lessons from one domain of medicine can guide the creation of principles for medical AI. Certainly Radiology is an area in which AI has been applied with some success, but this may cut both ways: success may appear more readily attainable than is likely to be the case (e.g. we have much further to go in the analysis of EHR data compared to image data).
3. Your phrasing suggests that these principles are necessary but not sufficient. It is important to be clear and explicit about this, including in your conclusions to reflect on how these principles may need to be supplemented or what more needs to be done.

Principle 1

4. As I read the title, I sense a dilemma: should we aim to make our AI tools equitable or should they be agents of "affirmative action," for want of a more precise term? I share your concern that AI may exacerbate inequalities, but I wonder if this is not something for the broader healthcare system to address. AI must address biases within its ranks and in its systems, of course, but does it somehow bear more responsibility than, say, oximetry, where it has been suggested that instruments perform less well on black skins? There is a case to be made here. Your guiding questions are the better pointers.
5. If it proved to be the case that DCIS is more prevalent in a particular racial group, but DCIS even there progresses to cancer in a minority of cases (say, 35% compared to overall 20% of cases), how should we strike an equitable balance, and how should AI help?

Principle 2

6. "For every disease state there are clearly defined, clinically meaningful outcomes." If only it were always so! But I have no quarrel with an imperative of clinically meaningful outcomes as the goal, recognizing that "no definite answer" is better than a confident wrong answer.

Principle 3

7. In the second line, "favoring sensitivity over specificity" would make the meaning clearer.
8. Overdiagnosis and overtreatment are undesirable, but so is misdiagnosis and delayed diagnosis. It seems to me that financial cost begins to figure very large in your thinking here and comes up time and again in the rest of the paper. Are there are other approaches than binary decision making? I don't know about DCIS in particular, but in prostate cancer "watchful waiting" or "active surveillance" provides a third way to deal with a very slow-growing tumor.

Principle 4

9. On arguments from cost: is it (always? ever?) possible to adopt a unidimensional measure of value in a multistakeholder environment? Your questions are again a better guide.
10. This is also the place to make the impact of opportunity cost on the overall system, as well as the benefit of early investment with long term pay-back.

Principle 5

11. “Biography” is a really nice term and deserves to be adopted, but “SDoH” or “SEDoH” are more immediately recognizable at present and would be worth mentioned.

12. The innocent-looking phrase “reasonably imputed” hides many dangers, including serious risk of bias. Perhaps this is what is special about AI, that every ingenious method we bring to bear on the solution of a problem brings its own danger in its wake.

Principle 6

13. This is a well-judged principle, and of course it is an area of active research.

Unfortunately, we know more negative cases (including adverse results) than significant successes. There should be a subdiscipline of transfer learning that deals exactly with what in an algorithm can be ported to another environment and what needs to be rethought each time.

Principle 7

14. You do not use the term, but in effect this is the principle of “mutability,” i.e., the approach to change, whether drift in the parameters of the system through “learning from experience,” through changes in the environment presenting “edge cases,” to rare unforeseen error through an undetected minor bug.

Conclusion

15. Is there a danger in displaying the seven principles in a way that makes them look like a checklist? There needs to be a message that these are necessary but not sufficient, they are sound as requirements, but not complete.

Reviewer #2 (Remarks to the Author):

1. Brief summary of the manuscript

The manuscript highlights the rise of clearances for AI-based tools in medicine and the initiatives committed to regulating and governing the approval and beneficence of such developments. The overall rationale is that there are fundamental lessons that have been learned in medicine that require careful consideration. The authors present seven principles targeted at ensuring this.

2. Overall impression of the work

While, on the surface, the work may appear derivative, given that several high ranking institutions have themselves formulated similar principles, the exposition strikes me as exceptionally clearly focused. It is following an approach that openly states significant issues without falling prey to the standard narratives that attest a general beneficence to the use of AI in medicine. This is refreshing and does not shy away from making very clear and – yes, debatable – suggestions, such as actively focusing on potential and already reported health disparities in conjunction with specific developments.

The way the authors exemplify their principles based on issues in AI-based breast cancer prediction is highly accessible and plausible.

The socio-economic focus that the principles take are a further novelty that is not sufficiently appreciated in the introductory paragraph.

3. Specific comments, with recommendations for addressing each comment

Overall, I believe that the authors should perhaps slightly work on the introductory paragraph to highlight the novelty and benefit of their approach to formulating principles. The socio-economic perspective is strong, as is the uncompromising attitude towards health equity (principle 1, 4 and 5).

However, I am missing one particular advancement in medicine that is not addressed and that I believe is profoundly endangered by AI, particularly those lacking explainability interfaces: patient-centric medicine involving shared decision making.

I suggest consulting the following two references for inspiration on that part. Perhaps an eighth principle could address this?

Bjerring, J. C., & Busch, J. (2021). Artificial Intelligence and Patient-Centered Decision-Making. *Philosophy & Technology*, 34(2), 349–371. <https://doi.org/10.1007/s13347-019-00391-6>

Herzog, C. (2022). On the Ethical and Epistemological Utility of Explicable AI in Medicine. *Philosophy & Technology*, 35(50), 31. <https://doi.org/10.1007/s13347-022-00546-y>

Reviewer #3 (Remarks to the Author):

General:

This viewpoint proposes seven principles that as the authors describe in their conclusion as “not... optional, but critical, and overdue”. The authors justify these principles based on the “lessons learned from medicine” which “must be systematically considered in the AI research process”. They provide useful examples about AI in their clinical field (related to breast cancer). If one looks first at the figure, p. 4, the explanation of the principles appears partly surprising: Why is this subgoal listed under the 6th principle “Designed to be population specific”? However, the link is convincingly explained in the text (p. 3): “Openly publishing AI workflows so institutions can train and evaluate their own local models is also needed.”

Critical remarks:

The way of writing is not always fully precise and arguments are not well elaborated due to the succinct style, but the authors’ claims are made sufficiently clear for the reader to understand them.

The most important lacking part seems to me an explanation why these principles are presented specifically related to AI. Indeed, the principles they establish concerning AI apply to any kind of medical technology, not only AI. It should be better explained why the seven principles are particularly important for AI. For example, this could be done referring to principle 6: only if algorithms are made OA can they be truly adapted to specific populations. Would similar specificities of AI make the other 6 principles also of particular importance for AI as compared to other technologies? Why not propose the seven principles to all medical technologies and interventions?

Why this could be problematic is shown by a subpoint of principle 6: “Will the AI workflow be made OA?”, should that be a criterion (for AI only, or all medical technology and interventions and why) or would that hamper the development of excellent AI because

developers lack a commercial incentive? At present algorithm (and medication or even Coca Cola) developers insist on the importance of IP, patents, as well as trade secrets. Another example is global equity (1st principle). This, as well as reduction of overtreatment (principle 3) and improvement of healthcare value (4.) concerns all health care technology, why require only AI to respect these principles? E.g.: “While some performance may be sacrificed to create equitable tools, realizing healthcare value for all is better than realizing healthcare value exceptionally well for the few” (Page 1 “1.” Line 12-14) – this is a particular view of justice and public health that is not shared by most capitalist and liberalist societies.

Similarly, p. 3, subheading 7.: “The FDA is also investigating whether companies should be allowed to submit a ‘change control plan’”. “We view this as a matter of necessity”. I agree with the authors on this point which, again, does not only apply to AI, but to most other medical technologies and procedures as the idea of learning health care systems applies to health care in general.

Conclusion: The seven principles go beyond the existing AI guidelines mentioned in the introduction and merit to be more widely discussed through publication in your journal. The viewpoint would profit from clarifications of the above mentioned issues.

Minor comments:

Page 1 “1.” Line 7: in the US (add “the”)

Page 2 “4.” Fourth line from bottom of paragraph: a “healthcare system” (add “a”)

Bernice Elger

Response to reviewers

Below, in purple bold text , is our responses to each comment by each reviewer. We thank the reviewers for taking the time out of their very busy schedules to review this manuscript and provide insightful comments.

Referee expertise:

Referee #1: PhD, AI, biomedical informatics

Referee #2: PhD, AI, ethics

Referee #3: MD, ethics

Reviewers' comments:

Reviewer #1 (Remarks to the Author):

Preamble

1. Thank you for the opportunity to review your principles for AI. You have entered a crowded field. Ted Shortliffe and colleagues are about to publish an entire volume on cognitive computation in biomedicine, and at least two professional societies, the AMA and AMIA, also offer definitions and principles, in addition to the organizations you mention. A little further afield, the IEEE has several standards working groups with remits that at least overlap medicine and healthcare.

Thank you for taking the time to review our work. We agree that several professional organizations have suggested AI principles. Thank you for drawing our attention to the AMA and AMIA principles. We have now included a table that summarizes principles from four organizations in this expanded Perspective piece. We did this to demonstrate how many of the proposed principles overlap and how, to our knowledge, some of our principles have not been explicitly stated in the literature. As we now state in the introduction, we believe that the value of our work is our emphasis on using AI to intentionally address our deeply engrained challenges in healthcare.

2. It is interesting to see to what extent lessons from one domain of medicine can guide the creation of principles for medical AI. Certainly Radiology is an area in which AI has been applied with some success, but this may cut both ways: success may appear more readily attainable than is likely to be the case (e.g. we have much further to go in the analysis of EHR data compared to image data).

We agree. The underlying questions, regardless of the AI application, are the same. The lessons learned in one application can be applied to another application. We also agree that some applications pose a more difficult challenge for AI as you have mentioned. Even with radiology AI, specifically mammographic AI tools, we see that there are the expected challenges with generalizability across different subpopulations.

3. Your phrasing suggests that these principles are necessary but not sufficient. It is important to be clear and explicit about this, including in your conclusions to reflect on how these principles may need to be supplemented or what more needs to be done.

Yes, you are right that these principles are not exhaustive. Other organisations have stated principles that we chose not to repeat which are now illustrated in Table 1. Further as the field of AI for evolves, we anticipate that further principles will be uncovered. We now explicitly state this in the introduction and conclusion.

Principle 1

4. As I read the title, I sense a dilemma: should we aim to make our AI tools equitable or should they be agents of “affirmative action,” for want of a more precise term? I share your concern that AI may exacerbate inequalities, but I wonder if this is not something for the broader healthcare system to address. AI must address biases within its ranks and in its systems, of course, but does it somehow bear more responsibility than, say, oximetry, where it has been suggested that instruments perform less well on black skins? There is a case to be made here. Your guiding questions are the better pointers.

This is a very thoughtful question on whether AI can be used as a tool of affirmative action. The application of affirmative action (AA) in healthcare has not really been explored. We found one article on this which we now cite under this principle. We also hesitated to phrase using AI as a form of affirmative action because AA is a challenging concept in the US, marred with much controversy. Nevertheless, we do briefly describe how AI can be used for AA.

5. If it proved to be the case that DCIS is more prevalent in a particular racial group, but DCIS even there progresses to cancer in a minority of cases (say, 35% compared to overall 20% of cases), how should we strike an equitable balance, and how should AI help?

AI should help by optimizing the prediction of who will and will not progress to invasive cancer with the consideration of subgroup analyses in your example. This could mean creating new models for ‘minority’ groups. More precise tools are generally needed; this is where AI can help.

Principle 2

6. “For every disease state there are clearly defined, clinically meaningful outcomes.” If only it were always so! But I have no quarrel with an imperative of clinically meaningful outcomes as the goal, recognizing that “no definite answer” is better than a confident wrong answer.

We agree. Thank you for this comment. We changed the language to be less definitive with an understanding that in some domains, what is a clinically meaningful outcome is not always clear.

Principle 3

7. In the second line, “favoring sensitivity over specificity” would make the meaning clearer.

Thank you for this comment. We edited this.

8. Overdiagnosis and overtreatment are undesirable, but so is misdiagnosis and delayed diagnosis. It seems to me that financial cost begins to figure very large in your thinking here and comes up time and again in the rest of the paper. Are there are other approaches than binary decision making? I don’t know about DCIS in particular, but in prostate cancer “watchful waiting” or “active surveillance” provides a third way to deal with a very slow-growing tumor.

We emphasize the issue of overdiagnosis and overtreatment because there is an overtreatment culture particularly in the USA. Yes, the financial costs is of significant concern to us, particularly in the USA where patients go bankrupt due to medical expenses. We are not clear where we propose a binary decision-making paradigm. We instead emphasize that the financial cost should be given more consideration than it presently is. Interestingly, our team is working on an active surveillance trial for DCIS. We have added this example under principle three.

Principle 4

9. On arguments from cost: is it (always? ever?) possible to adopt a unidimensional measure of value in a multistakeholder environment? Your questions are again a better guide.

We discuss here the healthcare value defined as the health outcomes achieved per dollar spent. The multistakeholder environment of healthcare – particularly in countries with public healthcare systems – is always limited by cost. It is an important, too often overlooked criteria. But this must also be considered alongside all of the principles proposed.

10. This is also the place to make the impact of opportunity cost on the overall system, as well as the benefit of early investment with long term pay-back.

Thank you for this comment. We were unsure how to integrate this into this piece and welcome any suggestions by the reviewer.

Principle 5

11. “Biography” is a really nice term and deserves to be adopted, but “SDoH” or “SEDoH” are more immediately recognizable at present and would be worth mentioned.

Thank you for this comment. Biography is encompasses more than SDoH, which we have clarified in the text.

12. The innocent-looking phrase “reasonably imputed” hides many dangers, including serious risk of bias. Perhaps this is what is special about AI, that every ingenious method we bring to bear on the solution of a problem brings its own danger in its wake.

Yes, imputation is a tricky strategy that can bring many risks. We have rephrased ‘imputed’ to ‘coded’ in using the example of geocoding to augment datasets. This was our original meaning.

Principle 6

13. This is a well-judged principle, and of course it is an area of active research. Unfortunately, we know more negative cases (including adverse results) than significant successes. There should a subdiscipline of transfer learning that deals exactly with what in an algorithm can be ported to another environment and what needs to be rethought each time.

This is an excellent suggestion, but we believe this point of transfer learning as a separate subdiscipline is beyond the scope of this paper.

Principle 7

14. You do not use the term, but in effect this is the principle of “mutability,” i.e., the approach to change, whether drift in the parameters of the system through “learning from experience,” through

changes in the environment presenting “edge cases,” to rare unforeseen error through an undetected minor bug.

Yes, mutability is a related term for dataset drift. To be consistent with the literature cited, we have changed the term to ‘dataset shift’.

Conclusion

15. Is there a danger in displaying the seven principles in a way that makes them look like a checklist? There needs to be a message that these are necessary but not sufficient, they are sound as requirements, but not complete.

We believe that a checklist is exactly what is needed. The many principles that AI teams must consider that traverse many disciplines including biomedicine, ethics, and law is daunting. We need comprehensive checklists that AI developers can use to make the process efficient. We have added in the introduction that these principles are not exhaustive and should be integrated with existing principles. We also explain in the introduction why checklists are needed.

Reviewer #2 (Remarks to the Author):

1. Brief summary of the manuscript

The manuscript highlights the rise of clearances for AI-based tools in medicine and the initiatives committed to regulating and governing the approval and beneficence of such developments. The overall rationale is that there are fundamental lessons that have been learned in medicine that require careful consideration. The authors present seven principles targeted at ensuring this.

Thank you for this excellent summary. This is exactly our point. The piece has expanded in length from a Comment to a Perspective piece. We hope that the additional text is further instructive.

2. Overall impression of the work

While, on the surface, the work may appear derivative, given that several high-ranking institutions have themselves formulated similar principles, the exposition strikes me as exceptionally clearly focused. It is following an approach that openly states significant issues without falling prey to the standard narratives that attest a general beneficence to the use of AI in medicine. This is refreshing and does not shy away from making very clear and – yes, debatable – suggestions, such as actively focusing on potential and already reported health disparities in conjunction with specific developments. The way the authors exemplify their principles based on issues in AI-based breast cancer prediction is highly accessible and plausible. The socio-economic focus that the principles take are a further novelty that is not sufficiently appreciated in the introductory paragraph.

Thank you for this comment. What you have described is what we hoped to accomplish – clear, focused guidance with an emphasis on fixing our existing mistakes in healthcare. One of our greatest challenges is our lack of emphasis on social and economic factors that drive care delivery and patient outcomes. Our introductory paragraph is now expanded to emphasize this as suggested.

3. Specific comments, with recommendations for addressing each comment

Overall, I believe that the authors should perhaps slightly work on the introductory paragraph to highlight the novelty and benefit of their approach to formulating principles. The socio-economic perspective is strong, as is the uncompromising attitude towards health equity (principle 1, 4 and 5). However, I am missing one particular advancement in medicine that is not addressed and that I believe is profoundly endangered by AI, particularly those lacking explainability interfaces: patient-centric medicine involving shared decision making.

I suggest consulting the following two references for inspiration on that part. Perhaps an eighth principle could address this?

Thank you for your insightful comments on our work. We have highlight in the expanded introduction the novelty and benefit of our work in the second paragraph of the introduction. When we submitted the Comment for review, we made note of the fact that we did not include a principle on why AI tools must consider patient values. This expanded Perspective piece allows for a larger word count therefore, we were able to include it as principle number eight.

Reviewer #3 (Remarks to the Author):

General:

This viewpoint proposes seven principles that as the authors describe in their conclusion as “not... optional, but critical, and overdue”. The authors justify these principles based on the “lessons learned from medicine” which “must be systematically considered in the AI research process”. They provide useful examples about AI in their clinical field (related to breast cancer). If one looks first at the figure, p. 4, the explanation of the principles appears partly surprising: Why is this subgoal listed under the 6th principle “Designed to be population specific”? However, the link is convincingly explained in the text (p. 3): “Openly publishing AI workflows so institutions can train and evaluate their own local models is also needed.”

Thank you for this comment. We were regrettably unable to integrate a suggestion based on this comment due to a lack of understanding as to whether there is a specific question or suggestion. We would welcome further clarification.

Critical remarks:

The way of writing is not always fully precise and arguments are not well elaborated due to the succinct style, but the authors’ claims are made sufficiently clear for the reader to understand them.

We were invited to resubmit this piece as a Perspective piece, which allowed for an increased word count. We therefore elaborated on most of the principles.

The most important lacking part seems to me an explanation why these principles are presented specifically related to AI. Indeed, the principles they establish concerning AI apply to any kind of medical technology, not only AI. It should be better explained why the seven principles are particularly important for AI. For example, this could be done referring to principle 6: only if algorithms are made OA can they be truly adapted to specific populations.

Would similar specificities of AI make the other 6 principles also of particular importance for AI as compared to other technologies? Why not propose the seven principles to all medical technologies and interventions?

We focused this piece on AI because of the nuance of this technology lends itself to unique considerations and for positioning these principles amongst the AI principle already proposed. We have clarified this in the second and third paragraphs of the introduction.

Why this could be problematic is shown by a subpoint of principle 6: “Will the AI workflow be made OA?”, should that be a criterion (for AI only, or all medical technology and interventions and why) or would that hamper the development of excellent AI because developers lack a commercial incentive? At present algorithm (and medication or even Coca Cola) developers insist on the importance of IP, patents, as well as trade secrets.

Yes, there is commercial incentive to create good AI however this doesn’t preclude companies from also creating open access tools. There are many examples of open AI tools including Google Collab and Tensor Flow to name two.

Another example is global equity (1st principle). This, as well as reduction of overtreatment (principle 3) and improvement of healthcare value (4.) concerns all health care technology, why require only AI to respect these principles? E.g.: “While some performance may be sacrificed to create equitable tools, realizing healthcare value for all is better than realizing healthcare value exceptionally well for the few” (Page 1 “1.” Line 12-14) – this is a particular view of justice and public health that is not shared by most capitalist and liberalist societies.

Similarly, p. 3, subheading 7.: “The FDA is also investigating whether companies should be allowed to submit a ‘change control plan’”. “We view this as a matter of necessity”. I agree with the authors on this point which, again, does not only apply to AI, but to most other medical technologies and procedures as the idea of learning health care systems applies to health care in general.

Conclusion: The seven principles go beyond the existing AI guidelines mentioned in the introduction and merit to be more widely discussed through publication in your journal. The viewpoint would profit from clarifications of the above-mentioned issues.

We respond to the previous three comments here which make the point that the principles discussed in our work could apply to any health technology. Yes, we agree that they could be applied to any technology. However, we have focused this perspective on AI for two reasons which we have now explained in the second and third paragraphs of the introduction. We are very grateful for your input.

Minor comments:

Page 1 “1.” Line 7: in the US (add “the”) **This was edited.**

Page 2 “4.” Fourth line from bottom of paragraph: a “healthcare system” (add “a”) **This was edited.**

Bernice Elger

REVIEWERS' COMMENTS:

Reviewer #1 (Remarks to the Author):

The authors have enhanced their submission considerably. They have included references to work that had been missed in the first draft and have addressed reviewers' comments to a large extent. What remains may be considered a "matter of taste" and should be allowed to stand.

Reviewer #2 (Remarks to the Author):

I believe that the authors have addressed all my criticisms and suggestions with care. The added length is warranted and I recommend the piece for publication as is.

Reviewer #3 (Remarks to the Author):

The authors have convincingly answered the reviewer comments and improved the manuscript.